# Fingerprint-Based Localization Approach for WSN Using Machine Learning Models

**Tareq Alhmiedat** [1,2]

---

1   Faculty of Computers and Information Technology, University of Tabuk, Tabuk 71491, Saudi Arabia;
    t.alhmiedat@ut.edu.sa
2   Artificial Intelligence and Sensing Technologies (AIST) Research Center, University of Tabuk,
    Tabuk 71491, Saudi Arabia

**Abstract:** The area of localization in wireless sensor networks (WSNs) has received considerable attention recently, driven by the need to develop an accurate localization system with the minimum cost and energy consumption possible. On the other hand, machine learning (ML) algorithms have been employed widely in several WSN-based applications (data gathering, clustering, energy-harvesting, and node localization) and showed an enhancement in the obtained results. In this paper, an efficient WSN-based fingerprinting localization system for indoor environments based on a low-cost sensor architecture, through establishing an indoor fingerprinting dataset and adopting four tailored ML models, is presented. The proposed system was validated by real experiments conducted in complex indoor environments with several obstacles and walls and achieves an efficient localization accuracy with an average of 1.4 m. In addition, through real experiments, we analyze and discuss the impact of reference point density on localization accuracy.

**Keywords:** fingerprinting; machine learning; indoor localization; received signal strength (RSS); range-free localization

## 1. Introduction

Recently, the quick evolution of embedded systems and radio waves has led to the advent of wireless sensor networks (WSNs), which have become a foremost research field in the last period. In general, a WSN is a collection of sensor nodes distributed over the area of interest to sense or monitor an event or set of events. A sensor node is an embedded device which consists of a transceiver, limited power resource, microcontroller, and array of sensors. Each sensor node can perform gathering and processing and communicate with nearby sensors. WSNs have been deployed widely in several applications, including industrial, military, environmental, home monitoring, and medical applications [1–4].

The positioning field has become an interesting field recently and has been adopted in several applications [5–7]. Localizing stationary and mobile targets in the area of WSNs is an interesting research field and involves estimating a location of a target object based on the existence of stationary sensor nodes distributed over the area of interest. Several WSN-based localization systems have been developed recently with different localization methods, accuracy, and costs. For instance, several types of measurements can be considered as position estimation methods, such as time difference of arrival (TDOA) [8], received signal strength (RSS) [9], time of arrival (TOA) [10], and angle of arrival (AOA) [11].

RSS-based localization systems offer reasonable localization accuracy outdoors; however, in indoor environments, the localization error becomes high due to the obstacles and walls that may exist, which usually weaken or strengthen the radio waves, hence increasing the localization error [12]. Mainly, WSN-based localization systems can be categorized into two main categories: triangulation and fingerprinting. The former triangulates the location of the target node using the RSS values received from stationary sensor nodes, whereas the

latter is based on collecting RSS values from stationary sensor nodes in a database and then estimates the position of the target node based on the stored RSS values.

On the other hand, the area of artificial intelligence (AI) represented by machine learning (ML) and deep learning (DL) techniques has been deployed widely in several WSN applications, including routing data, coverage problem, localization, and fault tolerance [13]. In this paper, we focus on the area of sensor node localization due to its significance in a wide range of applications, for instance, locating objects in a lab area, tracking patients in hospital, and localizing children in school.

There are several existing WSN-based localization methods and systems available to localize objects in indoor environments [14,15]; however, these systems are either high in cost, complicated, or inaccurate in complex environments. In addition, despite the available wide range of WSN-based localization using fingerprint systems, these systems are usually ineffective in real environments as most of the research studies focused on simulation experiments. Therefore, this paper discusses the research and development of an efficient localization system to accurately localize objects and items using the RSS technique and the investigation of several ML models. Hence, this project aims to:

1. Research the recently developed WSN-based fingerprinting localization approaches.
2. Construct a real RSS fingerprint dataset to allow researchers and developers to implement a real and efficient localization system for WSN application in indoor environments.
3. Develop an efficient device-free localization system using the RSS approach and tailored ML algorithms.
4. Investigate the impact of reference point density on the performance of localization accuracy.
5. Validate the efficiency of the developed system using real experiments conducted in the IIRC labs.

The rest of this paper is organized as follows: Section 2 discusses the recently developed fingerprinting localization systems, whereas in Section 3, the proposed fingerprinting localization system is presented and discussed. Section 4 discusses the experimental testbed in terms of the experiment testbed area, sensor nodes, communication protocol, and the collection of reference points procedure. In Section 5, the obtained results are analyzed and discussed, whereas Section 6 discusses the results obtained from real experiments and compares them with existing fingerprinting localization systems. And finally, Section 7 concludes the work presented in this paper and proposes future works.

## 2. Related Works

In general, WSN-based localization systems are categorized into two main categories: range-based and range-free techniques, as presented in Figure 1. The former requires an additional positioning device to be employed with each reference node (sensor nodes with known positions) or the target node (a sensor node with an unknown position), for instance, adding infrared, ultrasonic, and GPS-based methods [16,17], whereas the latter is based on the content of the transmitted message between the reference and the target nodes [18,19].

Range-free systems can be further divided into hop-count and multidimensional scaling (MDS) localization approaches [20–22]. Hop-count systems are based on the average hop distance between the reference node and target node, whereas the MDS systems estimate the target node's position based on the basic information of the reference nodes in the communication range. Both the hop-count and MDS systems reduce the hardware requirements for inexpensive sensor nodes; however, range-free approaches are inefficient in terms of localization accuracy.

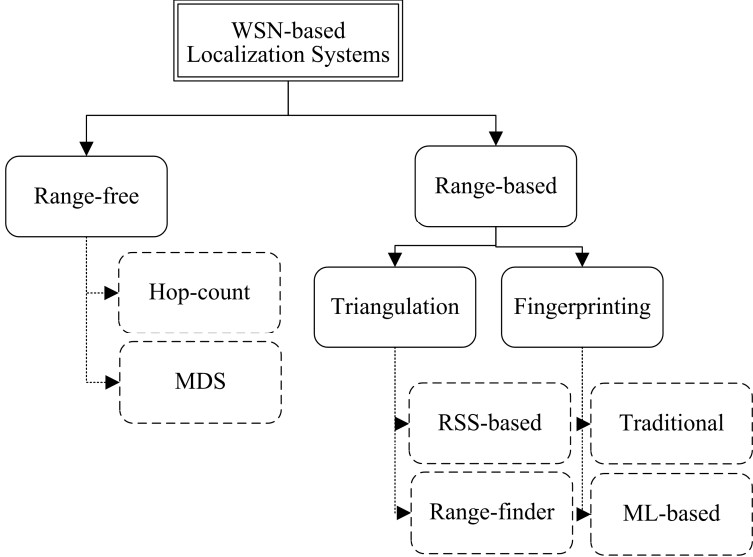

**Figure 1.** The categorization of WSN-based localization systems.

On the other hand, range-based localization systems have received considerable attention recently due to their efficiency in indoor environments and can be categorized into triangulation and fingerprinting localization systems. The former estimates the target node's location through triangulating the distance estimated from at least three reference nodes, whereas the latter is based on the behavior of signal propagation and information around the geometry of the tracking area.

The triangulation approaches can be categorized into received signal strength (RSS) and range-finder methods (such as infrared, ultrasonic, UWB, and RFID). In general, triangulation methods offer high localization accuracy in outdoor environments but fail in indoor environments, whereas fingerprinting achieves reasonable localization accuracy in indoor environments. Therefore, in this paper, we focus on employing a fingerprinting localization approach to track target nodes in the area of interest.

Fingerprinting approaches are categorized further into traditional approaches and artificial intelligence (AI)-based approaches. Both the traditional and AI-based fingerprinting localization approaches consist of two phases: the offline phase and the online phase. The main difference between the two approaches is the process of dealing with the fingerprints (RSS values with the corresponding 2D coordinates). The traditional approaches estimate the target node's position based on the nearest reference points collected in the offline phase, whereas the latter employed ML and DL approaches to train and estimate the target node's position.

Several WSN-based fingerprinting localization systems have been developed with various localization accuracy and efficiency [23,24]. However, these approaches offer unreliable localization information in complicated environments. Therefore, the AI-based approaches are considered in this paper. This section discusses the recently developed AI-based localization approaches that employed artificial neural networks and ML algorithms in the localization phase.

The authors of [25] proposed a hybrid target tracking system using the ML and Kalman filter to estimate the continuous location of the mobile target object. A fingerprint system involves the collection of RSS values from the tracking area, and then the authors employed an ML model for training purposes using the collected RSS dataset. Afterwards, the Kalman filter was employed to combine the predictions of the target's location based on the acceleration information with the first estimates.

The work presented in [26] involves developing a device-free wireless localization system using an artificial neural network. The developed system consists of two phases: the collection of the RSS values from the distributed reference nodes and the employment

of an ANN model to be trained using the collected RSS values. In the tracking phase, a non-linear function between the RSS inputs and outputs can be approximated using the pretrained ANN model.

In [27], the authors proposed a node localization system based on the Voronoi diagram and a support vector machine (SVM). The main idea is to divide the tracking region into several parts using the Voronoi diagram and anchor node in the localization region, and then the initial location of the target node is calculated through locating the region of the target node. Then, the SVM is employed to further optimize the position of the target node.

The authors of [28] proposed a novel algorithm that involves a DL model and high-level extracted features using autoencoder to enhance the localization accuracy. The authors raised the number of training data to improve the localization accuracy. On the other hand, the work presented in [29] involves an indoor localization system based on an SVM model. The authors employed a multi-class SVM with RSS measurements to develop a zoning localization approach. The work presented was validated in two different areas (hospital and laboratory buildings). The obtained results were compared with the ANN and showed the effectiveness of the presented SVM model.

In [30], the authors developed a WSN-based localization system using a cascaded layered recurrent neural network (L-RNN) for the classification of user localization in indoor environments. After considering several experiments by adopting different neural network models, the authors revealed that the experimental results showed that the implemented L-RNN model is an accurate localization method for indoor environments.

The work presented in [31] investigates the employment of ML algorithms for network-wide localization in large WSNs. The authors adopted an SVM model with a radio basis function (RBF) kernel for training the learning algorithm using the training dataset. The authors of [32] proposed a location identification method using the ANN method and RSS signals for sensor networks. The authors revealed that the performance analysis demonstrated the effectiveness of the proposed ANN model to estimate the location of the target nodes.

The authors of [33] proposed the employment of an outlier detection method for removing the effect of erroneous distance estimates in position estimating using the RSS method. In this work, the authors proposed three different localization schemes that apply the outlier detection to effectively minimize the localization errors in shadowed environments.

In [34], the authors proposed a range-free localization algorithm based on neural network ensembles (LNNEs), where the target's location is estimated using LNNEs solely based on the connectivity information in the WSN. The authors compared the obtained results with the centroid and DV-Hop range-free localization systems, where the experimental results demonstrated the efficiency of the LNNE approach.

As discussed above, various localization systems have been developed recently with different costs, accuracy, and experimental results. Table 1 presents a comparison among the existing AI-based positioning approaches for WSN, where the recently developed AI-based localization systems are discussed according to the following parameters:

1.  ML algorithm: localization data can be estimated using different types of ML algorithms. Therefore, it is important to determine the ML algorithm that has been employed in the offline and online phases.
2.  Experiment testbed: in general, experiments can be either simulation or real-time experiments. Simulation experiments are efficient in ideal situations; however, they offer unreliable results in complex environments. Real-time experiments, on the other hand, are hard to implement; however, they offer reliable localization results.
3.  Localization accuracy: localization systems need to be validated in order to assess the performance of the localization accuracy. Usually, localization accuracy is measured using the localization error in centimeters (cm) or meters (m).

**Table 1.** A comparison between the recently developed AI-based localization systems.

| Research Work | Algorithm | Experiment Testbed | Localization Accuracy |
|---|---|---|---|
| [25] | Kalman filter, ridge regression, and vector output | Simulation experiments | Localization error: 2 m |
| [26] | Artificial neural network (ANN) | Real experiments using the CC2530 ZigBee nodes | Localization error: 1.7 m |
| [27] | Voronoi diagram and support vector machine | Simulation experiments—MATLAB | Localization error: 0.3 m |
| [28] | Extreme learning machine | Simulation | Localization error: 2.5 m |
| [29] | Support vector machine | Simulation | Classification rate: 90% |
| [30] | Layered recurrent neural network | Simulation using real datasets | Accuracy: 93.55% |
| [31] | Support vector machine with radial basis function kernel | Simulation using real datasets | Localization error: 4 m |
| [32] | Artificial neural network (ANN) | Simulation experiments | Localization error: 6 m |
| [33] | Outlier detection | Simulation experiments | Localization error: 5 m |
| [34] | Neural network ensembles (LNNEs) | Simulation experiments | Localization error: 4.5 m |

As presented in Table 1, several AI-based fingerprinting approaches have been proposed recently with the aim of minimizing the localization error in WSN-indoor localization scenarios. However, most of the existing systems were tested using simulation environments, which might offer inefficient localization accuracy and limited flexibility. In addition, the issue of reference node density has a vital impact on localization accuracy, but this issue was not taken into consideration in the existing AI-based fingerprinting approaches. Therefore, the work presented in this paper overcomes the limitations that exist in the previous research by investigating the adoption of several ML models for the purpose of sensor node localization indoors, developing a real-time fingerprinting localization system using real sensor nodes and analyzes the impact of the density of reference points on localization accuracy.

## 3. Machine Learning-Based Localization System

In this section, the development of an efficient WSN-based localization system to position target objects in complex indoor environments is presented. The developed system consists of two main phases: offline and online. In the offline phase, the RSS values, along with the corresponding 2D coordinates, are collected from several reference points in the localization area and saved into a database file (csv file), and then, the collected RSS values are passed into an ML model for training purposes. Figure 2 shows the concept of the offline phase, which includes the collection process of RSS values from several reference points, preprocessing of the RSS values, and then performing the training of an ML process.

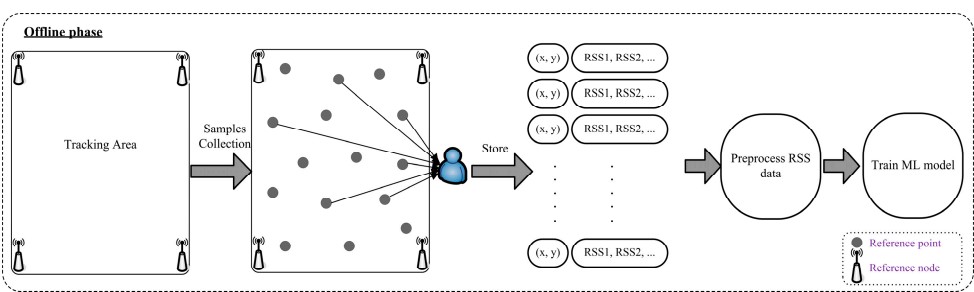

**Figure 2.** The offline phase, which includes gathering of the RSS values and the training process.

On the other hand, the online phase includes estimating the target node's position based on the RSS values received from reference nodes. Then, the collected RSS values are preprocessed and fed into a pretrained ML model to estimate the position of the target node. Figure 3 shows the concept of the online phase, which includes the collection of live

RSS values from the stationary reference nodes, processing the collected data, and then estimating the mobile target's 2D location (*x* and *y* coordinates) based on the pretrained ML model.

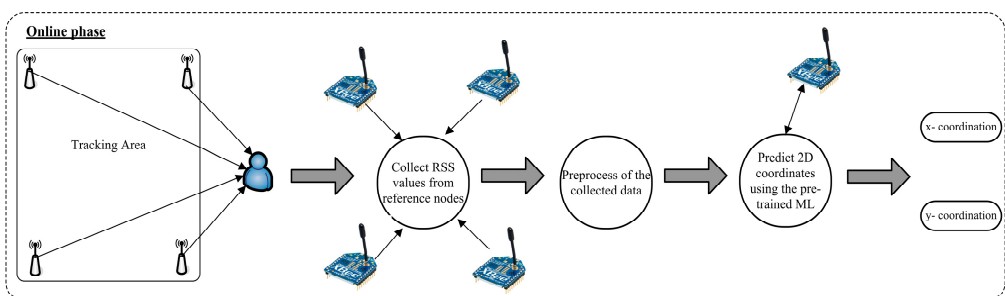

**Figure 3.** The online phase, which includes estimating the position of the target node.

The structure of the collected RSS dataset is presented in Figure 4, where the collected data consists of 6 attributes (4 features and 2 labels). The features set includes the RSS values from 4 different reference nodes, whereas the labels set is the corresponding location of the mobile target node.

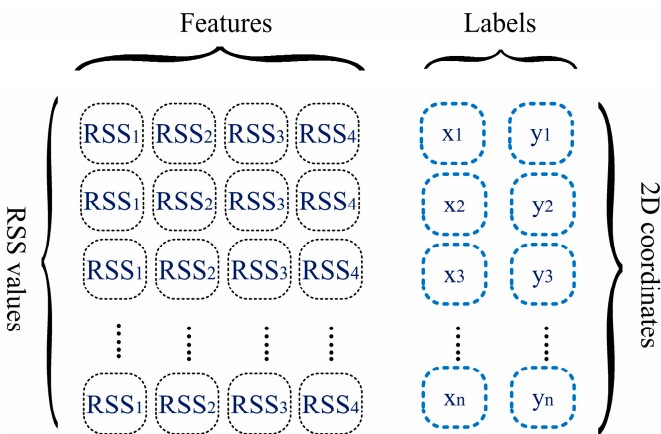

**Figure 4.** The structure of the RSS dataset.

The adoption of ML algorithms in fingerprinting localization systems improves the localization accuracy of the developed localization system in indoor environments [35]. In general, ML algorithms are designed to predict a single numerical value. However, some ML algorithms support multioutput regression. The presented work is based on predicting multioutput regression, which involves estimating two numerical values (x and y coordinates) for any target node location. Therefore, four ML models were tailored, tested, and adopted in order to enhance the localization accuracy for target nodes employed with a ZigBee communication protocol, as follows:

- Linear Regression (LR) is the most basic and commonly used category of predictive analysis, where LR investigates the relationship between one dependent variable and one or more independent variables.
- K-Nearest Neighbor (KNN) approximates the association between independent variables and the continuous outcome through averaging the observations in the similar neighborhood.
- Decision Tree (DT) is based on the form of a tree structure, where it breaks down the RSS dataset into smaller subsets, while an associated decision tree is incrementally developed.

- Random Forest (RF) employs the ensemble learning method for regression, where it combines predictions from multiple ML algorithms to make a more precise prediction than a single ML one.

The above four ML models were tailored to be suitable with RSS fingerprints through processing the RSS values and then estimating the localization information of the target node.

## 4. Experiment Testbed

This section discusses the experimental testbed in terms of experiment testbed area, the stationary reference nodes distributed in the tracking area, and the collection process of the RSS values in the offline phase.

### 4.1. Experimental Area

The experiments were conducted in the Industrial Innovation and Robotics Center (IIRC) lab at the University of Tabuk with the following dimension size (21.20 m × 7.60 m), as presented in Figure 5, where the IIRC lab includes different benches, devices, robots, equipment, and offices. As seen, there are walls and obstacles in the lab area, where radio waves may be weakened or strengthen accordingly. Figure 6 shows the IIRC lab layout with the 2D dimensions, whereas Table 2 presents the 2D coordinates for each reference node.

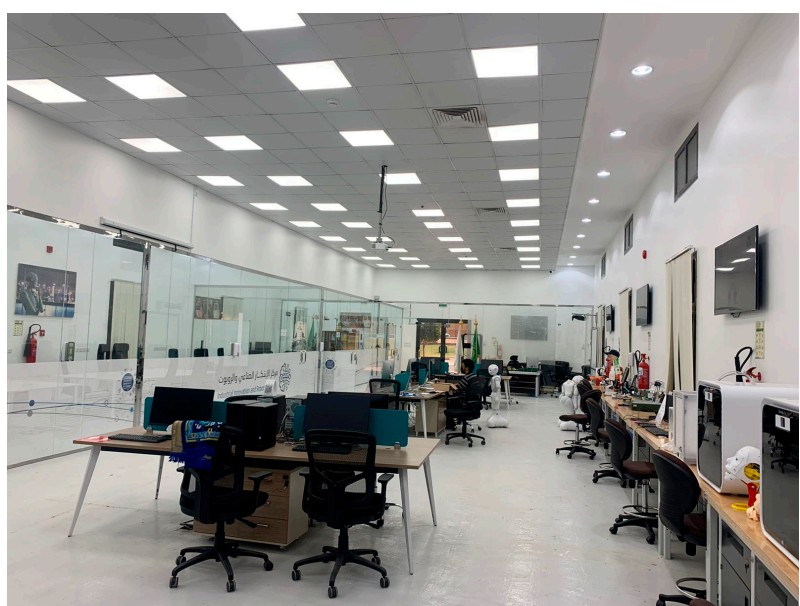

**Figure 5.** Photograph of the IIRC lab experimental scenario.

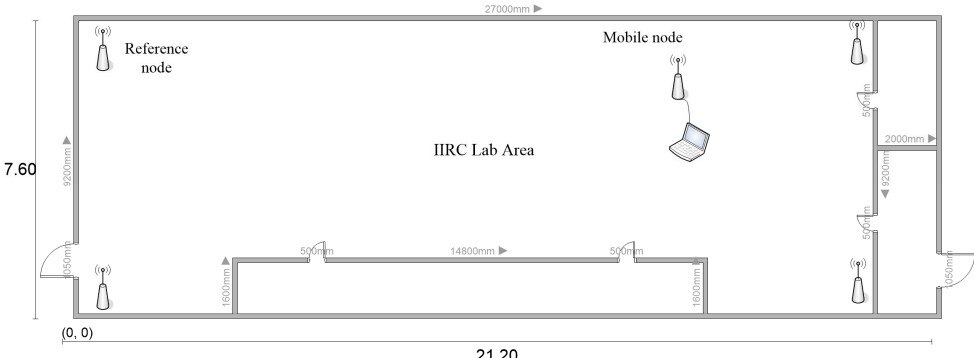

**Figure 6.** The layout of the IIRC lab with 2D dimensions.

**Table 2.** The 2D coordinates for each reference node.

| Node id | *x*-Cord (Meter) | *y*-Cord (Meter) |
|---------|------------------|------------------|
| Node 1  | 1.0              | 0.0              |
| Node 2  | 20.3             | 0.0              |
| Node 3  | 20.3             | 7.50             |
| Node 4  | 1.0              | 7.50             |

### 4.2. Reference and Mobile Nodes

The ZigBee communication protocol was employed in this study. In general, ZigBee is a low-data rate, low-power consumption, and low-cost wireless communication protocol. ZigBee protocol consists of 3 different type nodes: coordinator, router, and end-device. The experiment testbed consists of a ZigBee network with 5 sensor nodes (4 router and 1 coordinator nodes), as follows:

1. Stationary nodes (reference nodes): a number of 4-sensor nodes (router nodes) were placed in the corners of the IIRC lab at the University of Tabuk, as shown earlier in Figure 6, whereas Figure 8 depicts the developed ZigBee-based sensor node, which acts as stationary sensor node, and Figure 9 presents the architecture of the stationary sensor node.

2. Mobile node: a single mobile node (coordinator) is required to be employed to collect the transmitted frames from the stationary sensor nodes along with the RSS values for each received frame. Figure 7 shows the architecture of the mobile target node, which consists of an Arduino uno board to obtain and process the received RSS values from the distributed stationary reference nodes.

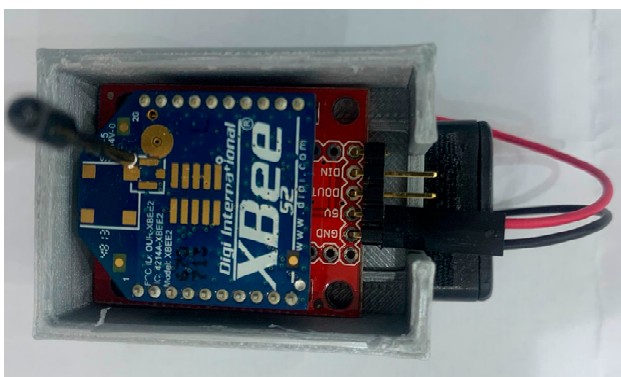

**Figure 7.** The architecture of the mobile target node.

Communication & control

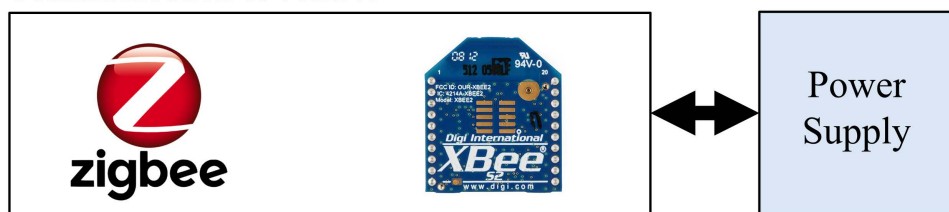

**Figure 8.** The developed stationary sensor node.

Controlling & Processing

**Figure 9.** The architecture of the station reference node.

The parameters for the experiment testbed are presented in Table 3.

**Table 3.** Experiment testbed parameters.

| Parameter | Value |
|---|---|
| Transceiver | XBee series 2 |
| Communication protocol | ZigBee |
| Transmission range | 100 m |
| Test-bed size | $20.2 \times 7.6$ m |
| number of reference nodes | 4 |
| number of target nodes | 1 |
| Power mode | 4: High |
| Node type | Router API |
| Minimum RSS value | 0 |
| Maximum RSS value | 100 |

*4.3. Collection of Reference Points*

This section discusses the collection of reference point phase. The reference points were collected manually from several points in the tracking area of interest and stored in a database file. Two different gathering processes (the collection of reference points in the tracking area) were conducted in order to analyze the impact of the reference point density on localization accuracy.

A total number of 68 and 126 reference points were collected from equally distributed points in the IIRC lab for gathering tasks 1 and 2, respectively. In addition, we merged the two datasets to produce a new RSS fingerprinting dataset with a total of 194 reference points. Table 4 presents general statistics on the 3 different RSS fingerprint datasets (small, medium, and large) collected. However, the difference between the RSS datasets is analyzed and compared according to the following parameters:

1. Total number of collected reference points (rp): this refers to the total number of reference points in the tracking area, where rp involves the 2D location of a reference point in the tracking area. Figure 10 depicts the total number of reference points that were collected using 3 different experiments (small, medium and large).
2. Density of reference points: this refers to the total number of reference points over the dimension (meter square) of the tracking area ($rp/m^2$).
3. Gathering process time: this presents the total time in minutes needed to accomplish the offline phase (the collection of reference points) for each RSS dataset.

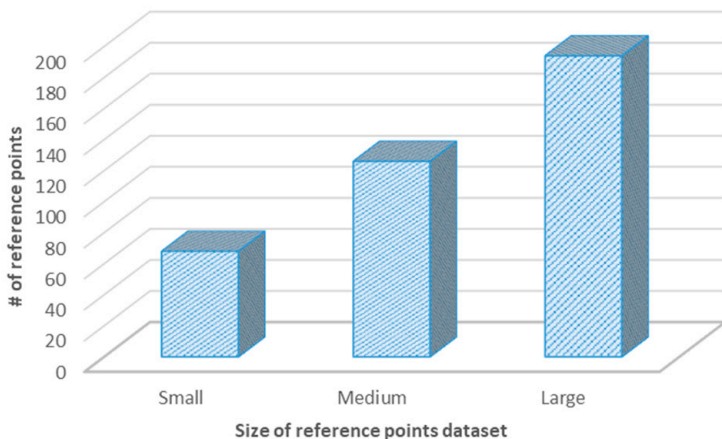

**Figure 10.** Total number of reference points for each dataset (small, medium, and large).

For validation purposes, the RSS datasets were divided into two subsets: 70% training subset and 30% testing subset, where the training subset is used to train the ML model, and the testing subset is employed to test the localization accuracy.

**Table 4.** General statistics on the RSS datasets.

| RSS Dataset | # of rp | Density (rp/m²) | Time | Training Size | Testing Size |
|---|---|---|---|---|---|
| Small dataset | 68 | 0.422 | 65 m | 47 | 21 |
| Medium dataset | 126 | 0.782 | 132 m | 88 | 38 |
| Large dataset | 194 | 1.216 | 197 m | 135 | 59 |

m: meter.

### 4.4. Hyperparameter Tuning for ML Models

A hyperparameter is a parameter of an ML model which is required to be set before starting the training process. Hyperparameter tuning is essential for any ML process, as a good choice of hyperparameters can make the model succeed in meeting the desired metric values. Therefore, this section presents the hyperparameter tuning for each ML model after considering several experiments. Table 5 presents the customized hyperparameter tuning for the LR model, where the number of splits was set to 10, and the number of repeats was set to 3. On the other hand, the customized hyperparameters for the KNN model is presented in Table 6, where the number of neighbors is equal to 5, and the metric distance function was set to Euclidian.

**Table 5.** Hyperparameter tuning for LR model.

| Parameter | Value |
|---|---|
| n—splits | 10 |
| n—repeats | 3 |
| Random state | 1 |

**Table 6.** Hyperparameter tuning for KNN model.

| Parameter | Value |
|---|---|
| n—neighbors | 5 |
| Metric | Euclidian |
| Sample weight | None |

The customized hyperparameter tuning for the DT model is shown in Table 7, where the max. depth value was set to 4, and the max. leaf-node value was set to 7. Finally, Table 8

presents the customized hyperparameter tuning for the RF model, with a max. depth of 4, n—estimators value of 1, and the min. sample split was set to 20.

**Table 7.** Hyperparameter tuning for DT model.

| Parameter | Value |
|---|---|
| Max. depth | 4 |
| Max. leaf nodes | 7 |

**Table 8.** Hyperparameter tuning for RF model.

| Parameter | Value |
|---|---|
| Max depth | 4 |
| n—estimators | 1 |
| Min. sample split | 20 |

## 5. Experimental Results

This section discusses the results obtained from employing four tailored ML models on three different RSS datasets with various density distributions of reference points. For evaluation purposes, we assessed the following metrics for each ML model:

1.  Mean absolute error (MAE): this refers to the magnitude in difference between the prediction of an observation and the actual value of that observation. The MAE is calculated as follows:

$$MAE = \frac{1}{N} \sum \left| Y - \hat{Y} \right| \tag{1}$$

    where $N$ refers to the total number of reference points (test points), $Y$ refers to the actual reference point, and $\hat{Y}$ refers to the estimated location.

2.  Standard deviation of MAE: this metric offers a general insight about the developed model. The MAE shows the performance of the ML model, whereas the standard deviation of the MAE shows how efficient the ML model is on the whole dataset.

3.  Average localization error (ALE): this refers to the difference between the predicted 2D coordinates and the real 2D coordinates ($x$ and $y$). Therefore, we estimated the localization accuracy for the estimated locations through calculating the difference between the estimated location ($x_e - y_e$) and the actual location ($x_r - y_r$), according to the following formula:

$$Loc_{Acc} = \sqrt{(x_e - x_r)^2 + (y_e - y_r)^2} \tag{2}$$

First, the MAE is assessed for the four ML models using the three different RSS fingerprint datasets. Figure 11 presents the MAE metric for the four ML models using the three datasets (small, medium, and large). As noticed, the LR model offers the best MAE results for the three RSS datasets with an average of 2.2 m, whereas the DT model offers almost the worst MAE results for the three RSS datasets. On the other hand, the RF and KNN models offer reasonable average MAE results compared to the LR model. The obtained detailed results for the tailored MAE model with each single dataset are shown in Table 9.

The KNN model offers the best MAE results with the large RSS dataset, whereas the RF model achieves the worst MAE score. The LR and DT models came in second and third place, respectively. As noticed below, the decision tree models achieve better MAE results compare to the LR models; this is because that the DT model supports non-linearity data values, such as the issue with the RSS fingerprint dataset. On the other hand, the KNN model achieves a much better MAE score than the LR one, as the KNN is a parametric model. The DT model is faster than the KNN model in real-time scenarios; however, the KNN model achieves a better MAE score compare to the DT model.

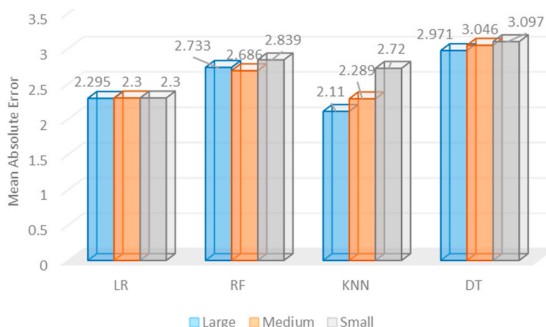

**Figure 11.** The MAE results for 4 localization methods using the 3 RSS datasets.

**Table 9.** MAE results for the 4 tailored ML models using 3 different datasets.

|  | Small Dataset | Medium Dataset | Large Dataset |
| --- | --- | --- | --- |
| LR | 2.300 | 2.300 | 2.295 |
| RF | 2.839 | 2.686 | 2.733 |
| KNN | 2.720 | 2.289 | 2.110 |
| DT | 3.097 | 3.046 | 2.971 |

Second, the standard deviation of the MAE was assessed. Figure 12 shows the standard deviation of MAE for the four ML models when adopted with the three different RSS fingerprint datasets. The large RSS fingerprint dataset offers a high mean absolute deviation for almost all of the four ML models, and this indicates that many of the predicted coordinate values are spread out further from the mean. On the other hand, the small RSS dataset offers the minimum standard deviation of the MAE for the four ML models, and this reveals that most of the predicted coordinate values are close to the mean, as the predicted distance from each coordinate value to the mean is small. In addition, Table 10 presents the detailed standard deviation of the MAE results for the four ML models with the three different datasets.

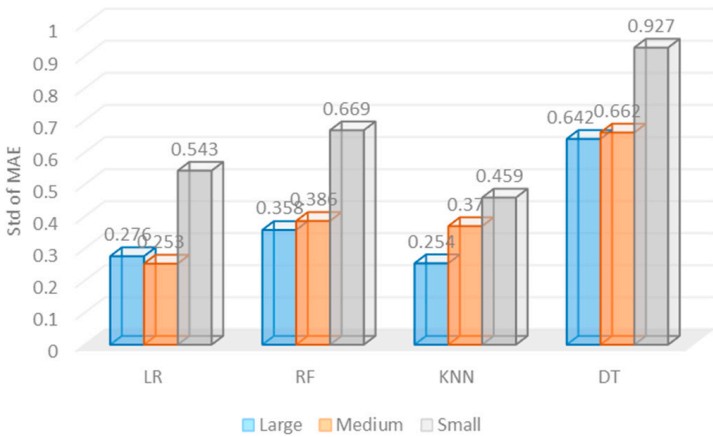

**Figure 12.** Standard deviation of MAE for 4 localization methods using the 3 RSS datasets.

**Table 10.** Standard deviation of MAE results for the 4 tailored ML models using 3 different datasets.

|  | Small Dataset | Medium Dataset | Large Dataset |
| --- | --- | --- | --- |
| LR | 0.543 | 0.253 | 0.276 |
| RF | 0.699 | 0.386 | 0.358 |
| KNN | 0.459 | 0.370 | 0.254 |
| DT | 0.927 | 0.662 | 0.642 |

Third, the localization error was assessed for each ML model using the three RSS fingerprint datasets. For instance, Figure 13 presents the average localization error for each ML model through the adoption of three different RSS fingerprint datasets. As presented below, the KNN model offers the best localization accuracy when adopted with the large RSS dataset, with an average localization error of (1.4 m). On the other hand, the RF model offers the worst localization accuracy with an average of (4.6 m) using the small RSS dataset, whereas the LR and RF models offer reasonable localization accuracy of 2.10 and 2.00 m, respectively, using the small RSS dataset.

However, according to the obtained localization accuracy, the large RSS dataset offers the best average localization accuracy compared to the small and medium RSS datasets, and this refers to the fact that the number of reference points is greater in the large RSS dataset, which assists the ML model to train on sufficient cases. Table 11 presents the detailed average localization error in meters when employing the four ML models with the three different datasets.

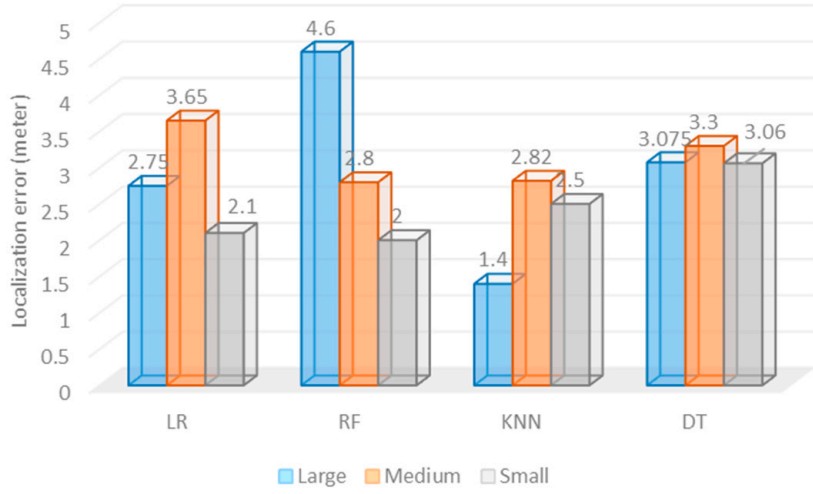

**Figure 13.** Average localization error for 4 localization methods using the 3 RSS datasets.

**Table 11.** Average localization error (in meters) for the 4 tailored ML models using 3 different datasets.

|  | Small Dataset | Medium Dataset | Large Dataset |
|---|---|---|---|
| LR | 2.10 | 3.65 | 2.75 |
| RF | 2.00 | 2.80 | 4.60 |
| KNN | 2.50 | 2.21 | 1.40 |
| DT | 3.06 | 3.30 | 3.08 |

For more evaluation analysis, the trained KNN model was tested in another indoor testbed that is presented in Figure 14, which is a study room located in the IIRC lab with a dimension of $14.1 \times 3.92$ m$^2$ and consists of a number of desks and chairs. A total number of four reference nodes were distributed in the corners of the study room, and a single mobile node (target node) was employed for the validation process.

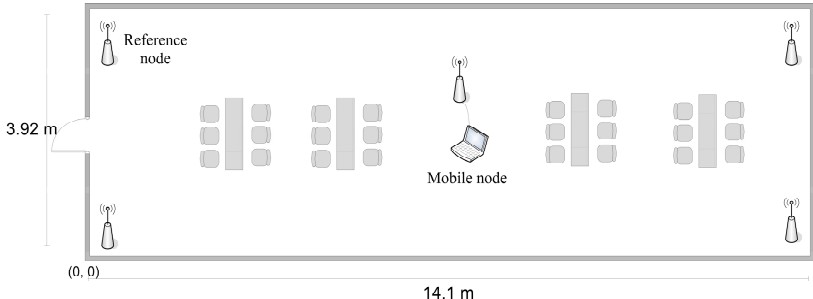

**Figure 14.** The actual structure of the study room.

For evaluation purposes, 20 different testing-points were positioned. Then, the RSS values were collected from each test point and fed into the tailored KNN model in order to perform localization estimation. The localization error was estimated through measuring the difference between the estimated and actual coordinates. As presented in Figure 15, the localization error was estimated for 20 test points, with an average localization error of 1.5 m.

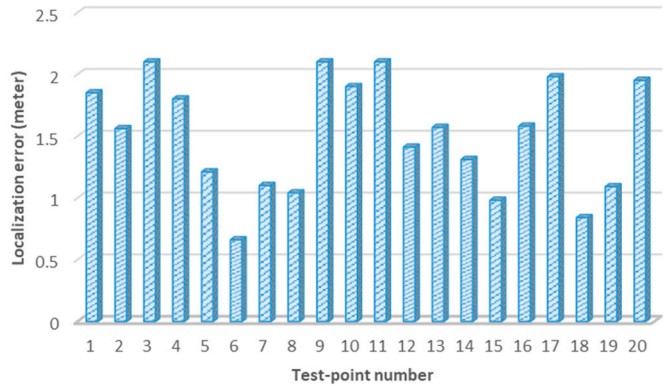

**Figure 15.** The localization error for 20 test points in the study room testbed.

On the other hand, the localization accuracy was analyzed for the developed system in this paper and the recent AI-based fingerprinting approaches. Table 12 presents a comparison between the developed fingerprinting localization approach that is based on the tailored KNN ML model and the recently developed ML-based fingerprinting localization approaches. As discussed above, most of the recently developed systems were validated through simulation experiments. In addition, the average localization error was 0.3–6 m. However, the developed system in this paper was practically validated through real experiments conducted in two different indoor environments, where the average localization error was around 1.4 m.

**Table 12.** A comparison between the developed fingerprinting localization approach and recent AI-based localization systems.

| Research Work | Experiment Testbed | Localization Error (Meter) |
|---|---|---|
| [25] | Simulation experiments | 2 m |
| [26] | Real experiments using the CC2530 ZigBee nodes | 1.7 m |
| [27] | Simulation experiments—MATLAB | 0.3 m |
| [28] | Simulation | 2.5 m |
| [31] | Simulation through real datasets | 4 m |
| [32] | Simulation experiments | 6 m |
| [33] | Simulation experiments | 5 m |
| [34] | Simulation experiments | 4.5 m |
| This work | Real experiments—ZigBee Series 2 nodes | 1.4 m |

## 6. Discussion

The radio frequency (RF) propagation channels in indoor WSN-deployment environments are commonly affected by shadowing due to the obstructions caused by natural and man-made obstacles. Therefore, triangulation-based localization systems offer efficient localization accuracy in outdoor environments (clean space) [36–38] but usually fail in indoor environments with the existence of walls and obstacles.

In general, RSS-based localization systems are easy to deploy, low in cost, and do not require complicated devices to achieve the positioning function. A device-free fingerprinting-based localization system is proposed in this paper with the investigation of adopting tailored ML algorithms to enhance the localization accuracy for indoor environments. Several research works considered RSS-based localization systems as range-based; however, in our approach, the RSS values are collected from stationary sensor nodes with no requirement of attaching additional sensors/devices to each reference node or mobile node [39]. Therefore, the developed system in this paper can be considered a range-free localization approach.

In this work, the positioning accuracy was further analyzed for both approaches, triangulation and fingerprinting. Therefore, two different experiments were conducted in the IIRC lab through implementing a triangulation approach to compare the localization accuracy with the obtained accuracy from the developed fingerprinting system. Through real experiments, the average localization error for the triangulation system was around (5.2 m); this refers to the structure of the IIRC lab, where the walls, obstacles, and objects existed. Figure 16 shows the localization accuracy results for 10 test points, conducted in the IIRC lab through deploying both the KNN and triangulation approach. As noticed, the KNN-based localization system achieves better localization accuracy than the traditional triangulation system in indoor environments.

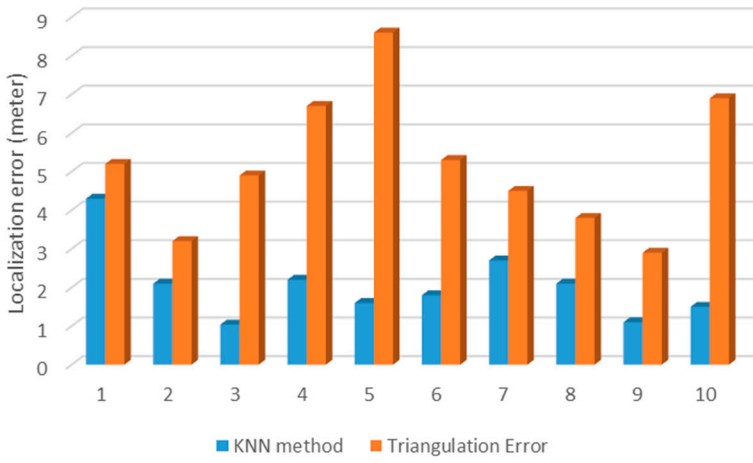

**Figure 16.** Localization error for KNN and triangulation systems in 10 different testing points.

On the other hand, there are several RSS–fingerprint datasets available online with diverse size and accuracy. For instance, the UJIINdoorLoc dataset [40], which consists of 21,048 records, covered three buildings, where the RSS values were collected from 520 different wireless access points (WAPs). The provided dataset is efficient for object localization using simulation studies. However, this dataset is inefficient in real experiments, as it requires a large number of reference nodes (WAPs) to be deployed in the area of interest, which is not valid in several localization scenarios. Therefore, this paper presents reliable and various RSS–fingerprint datasets with three different intensity levels of reference points, which are based on four reference nodes (four WAPs) and can benefit both simulation and practical studies.

Therefore, as noticed earlier, ML approaches have enhanced the localization accuracy of fingerprinting localization systems. Unlike the developed systems in [25,27,28,32–34]

which mainly focused on simulated experiments, we developed a real-time localization system using low-power consumption and low-cost sensor nodes (XBee Series 2 chips), where the developed system was tested in real indoor environments with different types of obstacles. In general, real experiments offer reliable and accurate results for the problems of indoor localization, as walls, obstacles, and dynamic objects affect the process of estimating the location of target nodes.

The presented research works in the literature [26,30,31] offer reasonable localization accuracy with an average of 1.7–6 m. However, the presented positioning system offers an accurate localization accuracy with 1.4 m, and this proved that the proposed fingerprinting localization system is efficient in indoor environments. Moreover, the developed stationary and mobile target nodes are simple, low in cost, and small in size, with no requirement of extra complicated sensor devices to accomplish the positioning phase, and can be attached to any object in a reliable manner.

As presented earlier in Table 3, increasing the number of reference points will increase the time required for manually gathering the RSS values from the allocated reference points. For instance, the time required to collect RSS values from 65 reference points is approximately 65 min, whereas 126 reference points requires 132 min, and finally the 194 reference points requires 197 min. Hence, collecting more reference points will increase the training and testing accuracy, increasing the localization accuracy. However, additional labor and cost are required to accomplish the offline phase, which may be unavailable in certain scenarios.

On the other hand, this paper investigates the impact of reference point density on localization accuracy. The results obtained from three different sizes of the RSS datasets with various reference point density values were analyzed and discussed. As a result, the medium and large density values offer high localization accuracy compared to the small RSS fingerprint dataset. However, the time required to accomplish the offline phase for the medium and large density values are longer than the small density environment.

### 7. Conclusions and Future Work

In this paper, the focus was on the field of WSN-based positioning systems using finger-printing and ML models, and four different ML models were investigated to accomplish the positioning task. As a result, the KNN model offered the best localization accuracy (1.4 m). In addition, the impact of reference point density on localization accuracy was investigated, and it was found that the environment with high reference points offers high localization accuracy. Moreover, three different real RSS fingerprint datasets were constructed in order to allow the researchers and developers to develop an efficient localization and tracking system for indoor WSN environments. For future works, the implementation of DL models will be considered, and several experiments will be conducted in different environments to improve the localization accuracy in indoor environments, and an autonomous robot system based on a robot operating system (ROS) will be employed in order to gather the reference points from the area of interest in a fast and reliable manner.

**Funding:** This research received no external funding.

**Informed Consent Statement:** Not applicable.

**Data Availability Statement:** Dataset available: https://www.kaggle.com/datasets/tareqalhmiedat/wifi-rss-fingerprint-dataset.

**Conflicts of Interest:** The authors declare no conflict of interest.

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
