# Peer review of "Fingerprint-Based Localization Approach for WSN Using Machine Learning Models"

_applsci, doi:10.3390/app13053037_

Round 1

Reviewer 1 Report

The paper is well-structured and well-written. There are a few suggestions and observations:

1. Figures 8 and 9 seem to have some color issues. The text in the blocks is not visible.

2. How can the gathering of reference points be made faster? A brief discussion may help other researchers to work on the problem.

3. As Figure 15 includes triangulation error on ten different points, this makes Figure 14 redundant. There is no need to keep Figure 14.

4. In Table 4, the unit used for Time is 'm', which makes it ambiguous. Whether it stands for milli or minutes, authors should mention minutes or what 'm' stands for. 

Reviewer 2 Report

The author proposes a Fingerprint-based Localization Approach for WSN using Machine learning models, and conducted comprehensive experiments to evaluate the accuracy. Overall, this paper is well organized and presented, there are some major suggestions focus on the experimental part:

1. Related work part: There are already many existing ML-based Fingerprinting Localization, the author needs to specify the contributions of this work by a single praragraph.

2. Page 6, "Therefore, four ML models have been adopted as follows" what is author's improvements for these models?

3. Experimental Results part: As the author discussed in the above parts that the proposed method is efficient, so the algorithm efficiency comparison is required in this part.

4. Only one experimental site is selected for evaluation, which is not enough. The reviewer suggests adding one more experimental site.

5. There are some public datasets which the author may consider to use, eg., IPIN2018, IPIN 2019

Round 2

Reviewer 2 Report

The author has addressed all my concerns, hence, I am glad to recommend this paper for publication in its current form.